# Concurrent AtC Multiscale Modeling of Material Coupled Thermo-Mechanical Behaviors: A Review

Yang Lu [1,*] , Stephen Thomas [2] and Tian Jie Zhang [3]

1   Civil Engineering Department, Boise State University, 1910 University Drive, Boise, ID 83725-2060, USA
2   Micron School of Material Science and Engineering, Boise State University, 1910 University Drive, Boise, ID 83725-1090, USA
3   Computer Science Department, Boise State University, 1910 University Drive, Boise, ID 83725-2055, USA
*   Correspondence: yanglufrank@boisestate.edu

**Abstract:** Advances in the field of processing and characterization of material behaviors are driving innovations in materials design at a nanoscale. Thus, it is demanding to develop physics-based computational methods that can advance the understanding of material Multiphysics behaviors from a bottom-up manner at a higher level of precision. Traditional computational modeling techniques such as finite element analysis (FE) and molecular dynamics (MD) fail to fully explain experimental observations at the nanoscale because of the inherent nature of each method. Concurrently coupled atomic to the continuum (AtC) multi-scale material models have the potential to meet the needs of nano-scale engineering. With the goal of representing atomistic details without explicitly treating every atom, the AtC coupling provides a framework to ensure that full atomistic detail is retained in regions of the problem while continuum assumptions reduce the computational demand. This review is intended to provide an on-demand review of the AtC methods for simulating thermo-mechanical behavior. Emphasis is given to the fundamental concepts necessary to understand several coupling methods that have been developed. Three methods that couple mechanical behavior, three methods that couple thermal behavior, and three methods that couple thermo-mechanical behavior is reviewed to provide an evolutionary perspective of the thermo-mechanical coupling methods.

**Keywords:** atomic to continuum coupling; concurrent coupling; thermal coupling; mechanical coupling; thermomechanical coupling

## 1. Introduction

Multi-scale material models generally refer to models that capture essential physics from disparate time and length scales and relate them. This coupled model can then result in simulations that explain and predict material properties and performances that cannot be captured in any single regime of the simulation. Multiscale methods enable computational material methods to challenge many of the fundamental limitations of continuum mechanics with larger atomistic simulations and sophisticated hybrid atomistic-continuum methods. One class of these multi-scale methods specializes in bridging between the atomic scale (angstrom) and the continuum scale (sub-micrometer). In this study, we focus on relating discrete atomic properties to their corresponding continuous property in the continuum formulation. Lower-scale models such as molecular dynamics (MD) and density functional theory (DFT) provide a relatively more accurate model of materials at high computational costs [1]. Higher scale constitutive models such as finite element analysis (FEA) or the phase field model (PFM) can efficiently model engineering scale structures and devices at relatively low computational costs [1]. However, these models must compromise accuracy when used to model a phenomenon occurring at smaller length scales because these methods assume homogeneity of material properties. Multi-scale models hold the promise of leveraging the best of both scales by combining the two used and coupled homogenization methods.

Apart from providing the ability to simulate larger length scales, multi-scale models are also useful in providing the framework for solving an atomic scale problem using continuum framework methods such as initial and boundary value problems [2]. Recent advances in characterization tools and processing methods have significantly increased our understanding of materials at the nanoscale, and this has led to the development of materials and devices engineered at that scale. This has, in part, driven the recent interest in multi-scale modeling.

An application area that can benefit from multi-scale models deals with exploring the structure–property relation in materials. For example, indentation tests are a common experimental technique used to determine the mechanical strength of materials. It is known that the continuum approximations of yield strength based on the Hall-Petch equation breakdown for grain sizes less than around 20 nm [3] and using MD to simulate even nano-indentation is prohibitively expensive, computationally. Miller et al. [4] recently used the quasi-continuum(QC) method to study dislocation nucleation in an fcc lattice during nanoindentation. The study revealed that the nucleus of the dislocation has a finite size that is linearly dependent only on the indent size and that the nucleation criteria need to consider the non-local nature of the atomic scale. Another application area that is well suited for multi-scale modeling is the study of the property and performance of engineered nano-scale materials, such as carbon nanotubes (CNT) and graphene. Continuum theory can be insufficient for studies involving a localized deformation causing dislocations or fractures since the continuum formulation assumes a homogeneous deformation. On the other hand, the MD approach is impractical when the application involves using CNTs in conjunction with other materials or devices which are larger and beyond the reach of MD. Qian et al. [5] used the bridging scale decomposition (BSD) method to study the post-bucking analysis of CNTs, where they used mesh-free approximations to represent the curved surfaces in the coarse scale. The coarse scale was used to impose essential boundary conditions, while MD provided fine-scale interatomic interactions. Nano-engineered devices are yet another application area that is an obvious target for multi-scale models given the small size scale inaccessible by continuum mechanics and complex geometries, which are hard to realize using MD alone. Templeton et al. [2] used spatially varying thermostats on a method based on BSD to emulate a nanodevice subjected to laser heating. The device consisted of a block of material with elastic properties of gold and a heat sink made of graphene-like material. This method utilized traditional FE boundary conditions, such as Dirichlet and Neumann boundary conditions to prescribe the temperature gradient and volumetric heat source with a Gaussian profile (laser heating), respectively, to the MD ensemble. Applying complex boundary conditions and initial conditions may be trivial for FE problems, but that is not the case for MD simulation, even though it is not immediately obvious. Multi-scale methods are making it easier to bridge this gap. For example, Templeton et al. [2] demonstrated the ability to prescribe heat flux to an FE-MD coupled system in arbitrary directions. This ability allowed them to estimate the anisotropic thermal conductivity of an atomic lattice with defects in equilibrium or non-equilibrium settings.

There are two distinct and important tasks in computational materials science research. The first involves the design and development of mathematical models that capture the essential physics required to explain material behavior. The second task is a simulation which refers to solving the equations that characterize the model for a particular physical system. The challenges of model development include developing insights into the physics of the system and interrelationships between them, whereas setting up simulations requires knowledge of numerical methods suitable for solving the governing equations in question, leveraging the computational power of modern supercomputers, and interpreting large amounts of data generated by the simulation.

The primary interest of this review is the former task of building material models. The fundamental law that describes material deformation behavior is Hooke's law and

is useful for describing elasticity in materials undergoing small deformation and being subjected to low loads (less than the yield stress). However, practical applications require more complex constitutive laws to be considered to accurately model material behaviors. For example, when a material exhibits different properties along different orientations, it may be necessary to consider using the Anisotropic linear elasticity model [6] instead of a linear isotropic model. An elastic perfectly plastic solid model [7,8] may be apt if the problem considers loads that exceed the yield stress, and the material being considered is a metal where it exhibits elastic behavior when loaded below the yield stress and deforms at constant stress beyond the yield stress. Such models use constitutive material parameters calculated from experiments as a dynamic input that depends on loading factors, such as pressure, moisture, and temperature. There are yet other models which require the explicit consideration of atomic-scale structures in the nanoscale because the material phase consists of nano-sized grains or is highly dependent on the dislocation dynamics or the material in the problem being considered is a single crystal. Such models include crystal plasticity [9] and discrete dislocation plasticity models [10] and often require the experimental values obtained from complex and expensive characterization tools. Other important models include models that describe interfaces between solids, models that describe a brittle fracture, fatigue crack growth, and stress corrosion cracking [11]. These models, which require the implicit consideration of atomic scale material structure, are, in general, more difficult to calibrate compared to the previous category because of the size scale involved and the limitations caused by the characterization techniques available.

An alternative approach is to replace the traditional stress-strain laws with direct calculations of stress-strain behavior using atomic scale simulations such as MD and molecular statics (MS). Typically, the atomic model is considered only in relatively small regions of the domain where its usage is imminent. The high computational cost involved in keeping track of a large number of degrees of freedom is the reason for limiting the spatial extent of the atomistic model. While in some cases, it may be advantageous to be able to model large spatial regions with atomistic refinement, in most cases, it is possible to assume homogeneous properties everywhere except in small regions with localized behavior that are different from the rest due to the atomic structure or physical loading conditions. For example, the mechanical properties of materials undergoing fracture can be assumed to be homogeneous everywhere except near the crack tip and the surfaces. Material models that mix the atomic and continuum models in the problem domain have naturally emerged from this nature of most physical material problems and are generally known as multi-scale models. A classic example of such a multi-scale model is the MAAD (macroscopic, atomistic, ab initio dynamics) method developed by Abraham et al. [12] to model the dynamic fracture of silicon.

This review is intended to provide a guideline for researchers beginning to explore this topic. The focus of this review is on concurrently coupling between MD and FEA. Other concurrent coupling hierarchies include coupling between DFT and MD [13,14], PFM, and FEA [15,16]. Typically, multi-scale modeling methods are categorized into two types: sequential and concurrent. Sequential methods, sometimes referred to as "Hierarchical" in the literature, are methods where the fine scale is simulated first, and results from it, usually constitutive parameters, are then fed as the input for the coarse scale. For example, Williamson et al. [15] calculated material properties such as thermal conductivity from molecular dynamics simulation and supplied it as the input for the continuum scale modeling of nuclear fuel irradiation effects to compensate for unavailable experimental data. On the other hand, in the concurrent methods, even though results from one scale may feed into another scale of simulation, this information passing happens at each time step and this information being passed is not necessarily constitutive of the parameters. For example, in the concurrent coupling of thermal transport developed by Templeton et al. [2], temperature, the primary variable, is solved for its exchange between the scales. The bridging scale decomposition (BSD) method [17] and the quasicontinuum

method [18] are typical examples of concurrent coupling methods. Concurrent methods can be further classified based on how the interface region between the atomistic domain and continuum domain is defined. Some methods [19] have a "handshake" region which is neither fully atomistic nor fully continuum in nature and provides a way to gradually transition between the atomistic and continuum models. There are other methods [17,18] that do not have a "handshake" region between the different models. Another feature that can easily distinguish the different methods is the coupling boundary condition which refers to whether the boundary atoms and finite element nodes are required to be coincident or not. These conditions are also referred [20] to as "strong compatibility" and "weak compatibility", respectively. The weak compatibility condition is procedurally simpler due to the reduced requirements for mesh refinement. Figure 1 is a pictorial depiction of the classification of multi-scale methods.

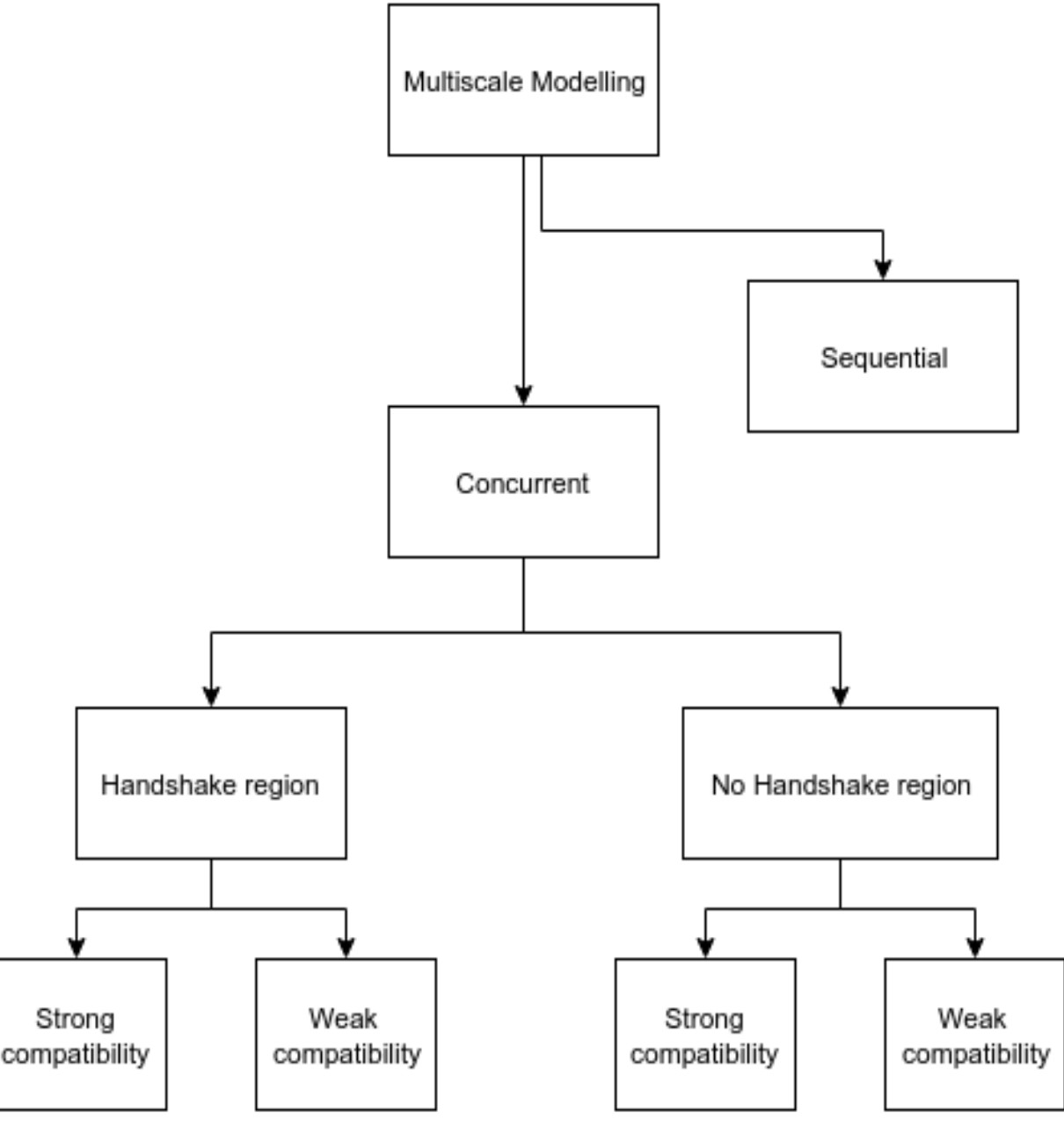

**Figure 1.** Classification of multi-scale models.

In typical multi-scale models, the coarse scale simulation is used in much of the problem domain. The designer of the simulation should carefully select the regions which should be simulated at a fine scale. This decision about the selection of the multi-scale boundary is one of the primary disadvantages of this method. However, some methods have overcome this issue by dynamically evolving the extent of the coupled region based on the variations in the state variables. For example, the QC method performs adaptive mesh refinement to achieve an atomic resolution in regions with a highly non-uniform deformation, such as dislocation cores and crystal boundaries. While some material phenomena that require multi-scale consideration are global in nature, others are applicable only in localized regions of the domain. Weinan [21] refers to this classification as type A and type B for the local and global problems, respectively. It is not difficult to see that some methods, such as QC, are more suitable for type B problems than other methods, such as the BSD method. Another important aspect of multi-scale modeling methods is how the scales are coupled temporally and not just spatially. The former is, in general, a more challenging task.

Given the scope of this paper, a good understanding of the fundamentals of MD and solid mechanics is important. A brief introduction to concepts related to MD is given here, and it should be considered as a list of pointers to topics that are relevant to discussions related to coupling MD with FE. The main goal of an MD simulation is to track the trajectories of a group of point particles representative of an ensemble of atoms while maintaining certain statistical properties which make the ensemble relevant to the calculation of thermodynamical properties. The trajectories of these particles are governed by Newton's second law of motion (Equation (1)).

$$\mathbf{f} = m\ddot{\boldsymbol{r}} \tag{1}$$

$$\mathbf{f} = -\nabla U \tag{2}$$

The force **f** is a known quantity given by the inter-atomic potential functions (Equation (2)) obtained either from quantum mechanical calculations, experimental data, or a combination of the two. The mass of the atom $m$ is also known, leaving the position $r$ as the only unknown quantity. Several time integration methods are available to solve the position of the particle as time progresses. Some of the popular methods are velocity-verlet [22], predictor-corrector [23], and r-RESPA [24] methods. Some qualities that differentiate the time integrators are their ability to explicitly include the velocity term as seen in the velocity-verlet equation (Equation (3)), the ability to handle noise in the force, and the ability to handle multiple time step dynamics. For example, the velocity-verlet method is better suited for ab-initio MD than the predictor-corrector method since the ab-initio MD experience has relatively more noise in the force than classical MD, which uses smoother empirical potential functions. Multiple time step dynamics are essential for multi-species ensembles where different molecules have different characteristic vibration frequencies, and therefore, a single time stepping scheme would be computationally wasteful since the lower frequency molecules do not need to be tracked at the same rate as that of the higher frequency molecules.

$$\boldsymbol{r}(t + \Delta t) = \boldsymbol{r}(t) + \mathbf{v}(t)\Delta t + \left(\frac{1}{2}\right)\mathbf{a}(t)\Delta t^2 \tag{3}$$

The subtle point to be noted in Equation (1) is that it is applicable for systems with only conservative forces acting on it. Mechanical forces are classified as internal, external, conservative, and non-conservative. The electrostatic force between a pair of monoatomic ions is considered as conservative because the force is only dependent on the instantaneous separation distance and not the atomic velocities. Forces, such as mechanical friction and viscous friction in gases and liquids, are considered non-conservative since the current magnitude of the force may depend on instantaneous velocities of atomic motion. When dealing with a non-conservative system, similar to a dissipative system where a damping

force exerted on a particle is proportional to its instantaneous velocity with the opposite sign, Newton's second law will take the form shown in Equation (4), where $\gamma$ is the damping constant.

$$m\ddot{\boldsymbol{r}} = \mathbf{f} - m\gamma\dot{\boldsymbol{r}} \tag{4}$$

Several practical issues are encountered while running MD simulations which is partly the motivation for coupling MD with continuum methods. Particularly, two issues are important to the discussions in this article. The first is the periodic boundary condition that is widely used in MD simulations to assume the infinite spatial extent of the material, and the second is related to the methods used to simulate NEMD using the concept of thermal exchange with an external heat bath [25].

The organization of the rest of this paper is outlined here. Section two describes a brief review of the important methods that have been devised to interpret atomic quantities in the continuum language. Section three reviews the coupling methods specifically implemented for mechanical deformation behavior, and section four reviews methods that couple thermal behavior. In section five, some of the existing methods for coupling thermo-mechanical physics in a multi-scale context are examined, and section six presents some conclusive thoughts about the methods reviewed.

## 2. Coarse-Graining Methods

In the continuum formulation, quantities are given by continuous functions that can be evaluated at any point in space. However, this is not the case with the atomic representation in which properties are defined only at the atomic position, and these properties need not be a function of just the single atom being considered. Hence the atomic representation is non-local in nature. In other words, atomic properties at a given point in space depend on their neighboring spaces, unlike the continuum representation, where properties are defined as continuous functions which can be evaluated at any point in space locally. This section describes some of the methods developed to address the need for translating non-local atomic properties to continuum representations.

### 2.1. Virial Theorem (VT)

The virial theorem (VT) was developed by Clausius [26] and extended by Maxwell [27] to define the virial stress as a tensorial quantity. The virial stress is basically the Cauchy stress in the language of atomic quantities. VT defines the isotropic pressure $P$ on an ensemble of $N$ atoms enclosing a volume $V$ as shown in Equation (5), where the first term involves temperature $T$ and the second term represents force due to the potential energy ($\varphi$) between two atoms $\alpha$ and $\beta$.

$$P = \frac{1}{V}\left(Nk_BT - \frac{1}{3}\left\langle\sum_{\alpha}\sum_{\beta}\frac{\partial\varphi_{\alpha\beta}}{\partial\mathbf{r}_{\alpha\beta}}\cdot\mathbf{r}_{\alpha\beta}\right\rangle\cdot\right) \tag{5}$$

Here, both terms on the right-hand side imply spatial and temporal averaging. $T$ implicitly implies temporal averaging, and the angled brackets on the second term explicitly imply temporal averaging. It is then necessary to perform this averaging over a subset of the ensemble, especially when the property is a field. These constraints restrict this method to be applied to the material models, which assume homogeneity and are in thermodynamic equilibrium. It has been shown that when applied to inhomogeneous and non-equilibrium conditions, the virial stress does produce errors [28,29]. Practical issues for MD simulation arise while calculating point-wise properties from such sub-ensembles, and they have been identified by Webb et al. [30]. However, due to its simplicity, it is widely used in MD studies for calculating pressure and virial stress.

## 2.2. Irving and Kirkwood Method (IK)

Irving and Kirkwood [31] presented one of the first efforts to interpret continuum scale quantities from microscopic quantities for non-equilibrium systems. They derived the equations of hydrodynamics from non-equilibrium classical statistical mechanics theory and defined expressions for density, velocity, and specific internal energy, the three independent variables in hydrodynamics using a probability density function. From these definitions, heat flux and stress tensor expressions were obtained, such that the balance laws of the continuum mechanics were satisfied. It is important to note that the densities obtained are pointwise functions which are phase averages and not macroscopic observables or continuum fields which require spatial and temporal averaging. This distinction has been clearly identified and emphasized recently by Admal et al. [32]. Obtaining these pointwise fields requires knowledge of the probability distribution function in phase space. However, in MD simulations, due to its purely deterministic nature, these probability density functions reduce to a Dirac delta function, and hence the pointwise fields are localized to particle positions. The Irving-Kirkwood method has three potential drawbacks. Firstly, the use of the probability distribution function meant that the method was stochastic in nature and led to practical difficulties due to the lack of knowledge of the probability distribution function. Secondly, this method used a series expansion of the Dirac delta distribution function, which was identified as mathematically non-rigorous by Walter Noll [33]. Lastly, the specific internal energy density derived in the Irving-Kirkwood method was shown [34] to be plausible only for identical particles interacting through pair potentials.

## 2.3. Hardy Method (HM)

Along the same lines as the basic idea introduced by VT and the Irving-Kirkwood method that discrete atomic properties can be related to continuous point-wise functions in the continuum formulation, the Hardy approach [35–37] is a notable method developed to model the mechanical stress introduced by shock waves simulated in MD simulation. The main contributions of the Hardy method were the introduction of a discrete and finite-ranged localization function which replaced the Dirac delta function in the Irving-Kirkwood method. Such functions, which provide a finite range, are also referred to as functions that provide compact support. It is a non-negative function that distributes the properties of an atom and allows all atoms to contribute to a continuum property evaluated at a position and time. Consider $\mathbf{X}$ as the position at which the continuum property is evaluated and $\mathbf{x}^\alpha$ as the position of an atom $\alpha$, then the localization function peaks at $\mathbf{x}^\alpha = \mathbf{X}$ and tends to zero as the distance between them becomes very large. The Hardy stress derived is shown in Equation (6). Here $\mathbf{x}^{\alpha\beta} = \mathbf{x}^\alpha - \mathbf{x}^\beta$, $\mathbf{F}^{\alpha\beta}$ represents the interatomic force between the particles $\alpha$ and $\beta$, the bond function $B^{\alpha\beta}(\mathbf{x})$ defined in Equation (8) can be interpreted as the percent of the bond between atoms $\alpha$ and $\beta$ that resides in the characteristic volume around the material point $\mathbf{x}$. The localization function $\psi$ given in Equation (9) can be interpreted as the percent of atom $\alpha$ that resides in the characteristic volume. In other words, the bond function accounts for the partial contribution of an atomic bond to a characteristic volume being considered. It is important to note that the kinetic term uses a velocity ($\mathbf{u}^\alpha$) relative to the continuum spatial point to access a kinetic contribution to the stress tensor instead of using an absolute atomic velocity. The decomposition of the absolute atomic velocity $\mathbf{v}^\alpha$ into the continuum velocity $\mathbf{v}(\mathbf{x}, t)$ and a relative velocity $\mathbf{u}^\alpha(\mathbf{x}, t)$, according to Hardy, is given in Equation (7). Such considerations make it different from virial stress, even though it looks very similar.

$$\boldsymbol{\sigma}(\mathbf{x}, t) = -\left\{ \frac{1}{2} \sum_{\alpha=1}^{N} \sum_{\beta \neq \alpha}^{N} \mathbf{x}^{\alpha\beta} \otimes \mathbf{F}^{\alpha\beta} B^{\alpha\beta}(\mathbf{x}) \right.$$
$$\left. + \sum_{\alpha=1}^{N} m^\alpha \mathbf{u}^\alpha \otimes \mathbf{u}^\alpha \psi(\mathbf{x}^\alpha - \mathbf{x}) \right\} \tag{6}$$

$$\mathbf{v}^\alpha = \mathbf{v}(\mathbf{x}, t) + \mathbf{u}^\alpha(\mathbf{x}, t) \tag{7}$$

$$\psi(\mathbf{x}^\alpha - \mathbf{x}) - \psi\left(\mathbf{x}^\beta - \mathbf{x}\right) = -\mathbf{x}^{\alpha\beta} . \partial_\mathbf{x} B^{\alpha\beta}(\mathbf{x}) \tag{8}$$

$$\int \int \int_{R^3} \psi(\mathbf{x}^\alpha - \mathbf{x}) d^3 \mathbf{r} = 1 \tag{9}$$

Hardy's method makes assumptions of force and potential energy, which are simple and are clearly valid only for simple pair potentials. For example, the potential energy of an atom was assumed to be a function of the distance between itself and the other atoms individually. Though this assumption agrees with pair potentials, such as Lennard Jones and the embedded atom model (EAM), it is invalid for 3-body potentials such as the Stillinger–Weber potential commonly used for modeling silicon. This has been identified by Webb et al. [30] and Zimmerman et al. [38]. Zimmerman et al. compared VT stress with Hardy stress and found that Hardy provides an equal to or more accurate definition of the Cauchy stress in an atomic system. They also found that the fluctuations in Hardy stress reduce in magnitude with the use of a smooth, continuous localization function with compact support. Zimmerman et al. [39] showed in another work that the problem of separating the potential and kinetic terms in the virial stress can be avoided by using a formulation similar to the Hardy method in the Lagrangian or material frame instead of the Eulerian or spatial frame. It was shown that the derived first Piola–Kirchhoff stress is consistent with the Cauchy stress calculated using the Hardy method.

### 2.4. Atomic Kinetic Temperature

In the MD simulation of classical systems, the simplistic definition of the temperature of an ensemble of N atoms in the thermodynamic equilibrium refers to the atomic kinetic temperature, which is given by Equation (10) where $m^\alpha$ is the mass of the atom $\alpha$, v is the velocity, and $k_B$ is the Boltzmann's constant.

$$T(t) = \frac{1}{3Nk_B} \sum_{\alpha=1}^N m^\alpha (v^\alpha)^2 \tag{10}$$

This temperature can be thought of as the vibrational energy of atoms. This notion holds good for systems where the atoms vibrate with a narrow vibrational spectrum. For example, a solid in equilibrium with a heat bath has atoms that vibrate at frequencies that are characteristic of the equilibrium thermodynamic temperature. However, if this solid is subjected to mechanical deformation, the atoms undergo vibrations of lower frequencies that are typical of the deformation motion, and then the simplistic definition of the temperature becomes insufficient. In other words, when a system is not in thermodynamic equilibrium, the kinetic temperature is not valid. The correct definition of non-equilibrium temperature is an area of active research. Some other definitions of temperature include the Metropolis Monte Carlo (MC) temperature, Rugh's temperature, and configurational temperature. A good discussion on this topic is provided by Powles et al. [40].

### 2.5. Hardy Temperature (HT)

A very similar measure of temperature was derived by Hardy et al. [37], as shown in Equation (11), where relative velocity ($\mathbf{u}^\alpha$) was used instead of absolute atomic velocities. The decomposition of the absolute atomic velocity is shown in Equation (7), and the localization function $\psi$, also known as the bond function, is given in Equation (8).

$$T(\mathbf{x}, t) = \frac{1}{3k_B} \frac{\sum_{\alpha=1}^N m^\alpha (\mathbf{u}^\alpha)^2 \psi(\mathbf{x}^\alpha - \mathbf{x})}{\sum_{\alpha=1}^N \psi(\mathbf{x}^\alpha - \mathbf{x})} \tag{11}$$

Webb et al. [30] observed that this method underestimated temperature relative to the temperature measured using the discrete method (Equation (10)) when the spatial

averaging sphere radius was small ($R_a = 0.5$ nm) but agreed with the discrete method, which had a sufficiently large spatial averaging ($R_a \gtrsim 1.5$ nm). Based on this observation, they concluded that it was best to use the Hardy method for interpreting stress and not temperature. However, it must be noted that the temperature equation was not rigorously derived by Hardy, as it was for the Hardy stress and heat flux, and this was noted by Webb et al. Another important observation by Webb et al. was that both discrete and Hardy methods for measuring heat flux in an atomic ensemble demanded larger degrees of temporal averaging ($t_a \gtrsim 100$ ps) to obtain fewer fluctuating values compared to the measurements of stress and temperature. This result was attributed to the inherent nature of these properties.

### 2.6. Equivalent Continuum (EC)

A purely mechanical theory derived by Zhou [41] that defines work-conjugate stress and deformation fields from non-local MD system. The discrete particle systems considered exhibit micropolar interatomic interactions which involve both central interatomic forces and interatomic moments. The equivalence of the continuum to discrete atomic systems includes preservation of momenta, conservation of work rates, and conservation of mass. The equivalence holds for the entire system and for volume elements defined by any subset of particles in the system. EC provides an explicit account of arbitrary atom arrangement, admitting applications to both crystalline and amorphous structures.

## 3. Downscaling Methods

*Thermostats*

Classical MD simulations produce a system whose total energy is constant, in other words, an isolated system. This type of ensemble is also known as a microcanonical or an NVE ensemble. However, most physical systems are kept at a constant temperature during experiments, and hence, it is convenient to be able to maintain the MD ensemble at a constant temperature rather than constant energy for comparison with experiments. Such an ensemble is known as the canonical or NVT ensemble. Several classes of thermostats have been developed over the years to tackle the problem of maintaining the desired temperature in a system of particles, and these thermostats are, in general, referred to as isokinetic thermostats. Some useful qualities of thermostats, in general, include time reversibility, the ability to achieve the Maxwell–Boltzmann distribution of velocity, the flexibility to prescribe the type of temperature and be able to promote heat flow. Figure 2 shows the three major classes of thermostats and examples for each class.

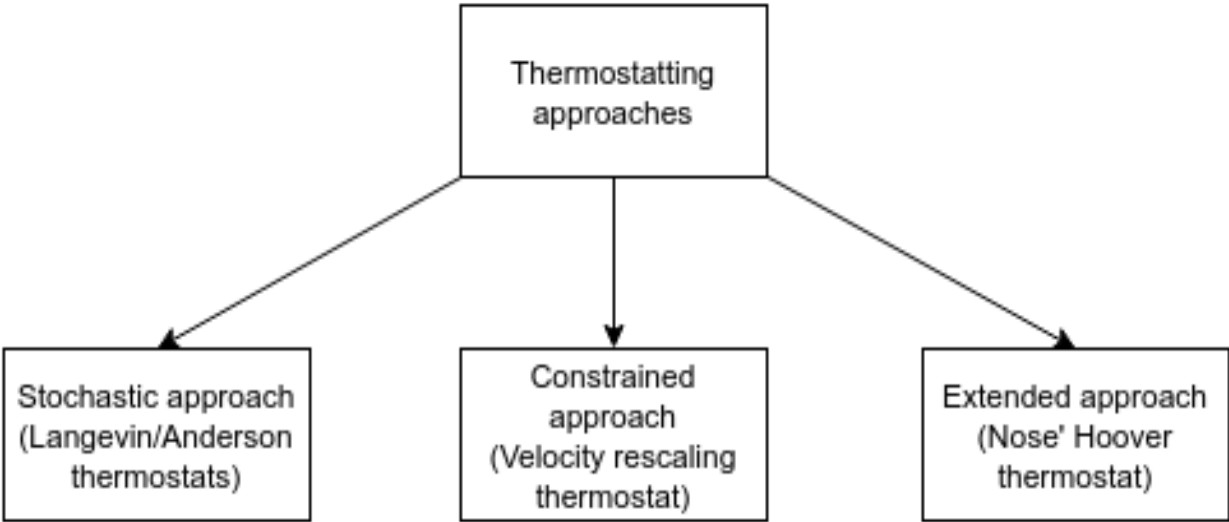

**Figure 2.** Classification of thermostats.

The velocity rescaling [42] thermostat is, in general, the least accurate since it lacks the ability to produce the thermal fluctuations typical of the canonical ensemble. However, this thermostat is quite efficient and, hence, is used mostly to perform the equilibration of an MD ensemble before the actual production simulation is performed. The Langevin [43] and Nose' Hoover [44] thermostats are able to reproduce the canonical ensemble, and these two types of thermostats are also referred to as Gaussian thermostats or Gaussian isokinetic thermostats because the equations of motion for these thermostats can be derived from Gauss' principle of least constraint [45]. The equations of motion for the atoms subjected to the Langevin and Nose' Hoover thermostat are shown in Equations (12) and (13), respectively. Here the $\gamma$ is similar to the damping coefficient seen in Equation (4) and G(t) in Equation (13) is the stochastic component. The Nose' Hoover thermostat is time-reversible, but the Langevin thermostat is not time reversible and can produce non-physical dynamics when velocities or positions become discontinuous in the phase space. This discontinuity in phase space happens because the additional momentum applied to the random atoms is chosen randomly from a Maxwell–Boltzmann distribution for the desired temperature.

$$m^{\alpha}\ddot{\mathbf{r}}^{\alpha} = \mathbf{f}^{\alpha} - m^{\alpha}\gamma\dot{\mathbf{r}}^{\alpha} \tag{12}$$

$$m^{\alpha}\ddot{\mathbf{r}}^{\alpha} = \mathbf{f}^{\alpha} - m^{\alpha}\gamma\dot{\mathbf{r}}^{\alpha} + \mathbf{G}^{\alpha}(t) \tag{13}$$

Unlike the continuum interpretation of stress where the atomic positions were sufficient, temperature and heat flux expressions are inherently associated with atomic motions and, therefore, need to consider the temporal variation and averages. As pointed out by Webb et al. [30], the two critical parameters that need to be configured for measuring the temperature of an atomic ensemble are spatial and temporal averaging. For temporal averaging, they identified that the sampling frequency and total analysis time were important quantities to be considered. Keeping the sampling frequency fixed at a conservatively high value of once every ten-time steps (the typical time step in MD is 1 fs), they studied the effect of the total analysis time and spatial resolution on the fluctuation in temperature measured using Equation (10). It was observed that a higher spatial resolution lowered the demand on the temporal resolution and vice versa. As a general guideline based on the specific MD simulation they used, it was found that a good balance of spatial and temporal resolution values was from 1 nm to 1.5 nm for the sampling sphere radius and from 5 ps to 10 ps for the total analysis time ($t_a$). However, no quantitative relationship between spatial and temporal resolution was derived. With the recent progress using machine learning methods, a unified thermodynamics-informed neural network has been developed to accelerate conventional iterative flash calculation schemes [46]. The thermodynamics-informed neural network provides a fast, accurate, and robust approach to calculating the phase equilibrium properties for unconventional reservoirs [47].

## 4. Mechanical Coupling Methods

### 4.1. Quasicontinuum Method (QC)

The QC method, originally developed by Tadmor et al. [18,48], can be simply thought of as an atomistic-based continuum method, hence the name Quasicontinuum. This method takes advantage of the fact that the material parameters that characterize the constitutive relation between the stress and strain used in FEA are, in many cases, the direct result of microscopic processes at the atomic scale. The QC method draws the constitutive input directly from atomistic calculations instead of using experimentally derived parameters. These constitutive parameters are input into the FEA model at quadrature points within the element. The strain energy density is calculated at the quadrature point by first selecting a representative atom or rep atom, which is surrounded by a small collection of atoms referred to as a crystallite with a radius $R_c$. In its simplest form, the relation between the continuum scale stress-energy density and the deformed crystallite is based on the Cauchy–Born rule (CB). The distribution of the

stress-energy within the rest of the element from the quadrature point is performed using interpolation functions. The deformation of the crystallite is driven by the local continuum deformation field evaluated at the quadrature point. The deformation is first applied to the basis vectors, and the crystal structure is reconstructed from the deformed basis. In the case where the crystallites do not overlap, the atoms in it only experience homogeneous deformation and can be thought of as an infinite crystal undergoing uniform deformation. This is referred to as the local QC formulation. Even though this assumption is elegant and perfectly suited for the definition of Cauchy stress, the deformation field is local in nature and is not representative of actual atomic systems. The local QC formulation is not capable of modeling inhomogeneous deformation processes, such as stacking faults, free surfaces, or grain boundaries. To enable non-local behavior, a non-local QC formulation has also been developed in which the element size can be smaller than the representative crystallite radius $R_c$, such as that of atoms in a crystallite experience, the influence of nearby crystallites as shown in Figure 3.

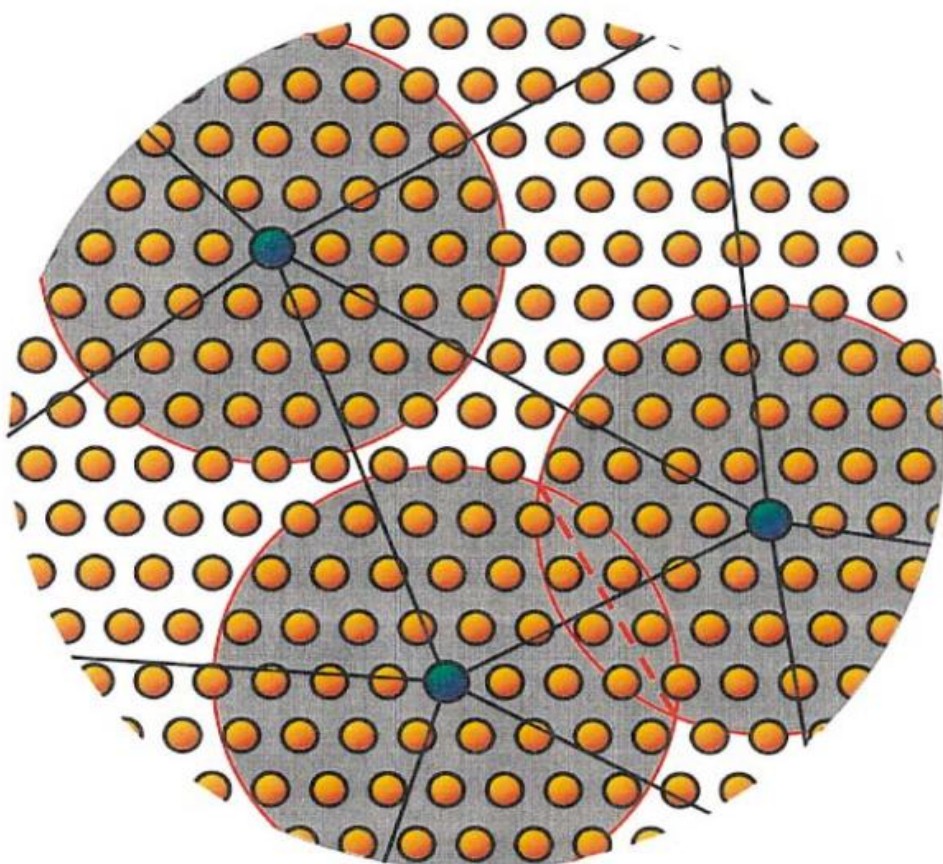

**Figure 3.** Illustration of non-local QC formulation with overlapping crystallites. Reproduced with permission from [49].

The atoms are assigned to a crystallite based on the closest rep atom, and if an atom is equidistant from two rep atoms, the assignment is random. The size of the crystallite on the atomic scale and the density of rep atoms in the continuum scale are two parameters that can be controlled in such a way that when the deformation gradient is steep, the mesh can be refined to the limit where the nodal degrees of freedom equal the atomic degrees of freedom. On the other hand, when in regions where the fields vary slowly, the mesh can be coarsened since it is unnecessary to track all the atomic degrees of freedom. A full three-dimensional version of the QC method was presented by Ortiz et al. [49]. The original QC method is also known as the "cold" QC because

molecular statics calculations at zero Kelvin are performed at the atomic scale. The finite temperature QC method, also known as "hot" QC, was subsequently developed and several authors have contributed to this idea, thereafter [50–52]. Furthermore, "hot" QC "dynamic" methods have been developed where the atomic and continuum regions evolve dynamically and in tandem [53]. This method was found to be computationally more efficient. One of the drawbacks of the QC method is the problem of spurious wave reflections at the atomic/continuum boundary where high-frequency phonons either generate due to thermal motion or by other atomic-scale phenomena can move freely through the atomic scale but cannot be represented in the shape functions in the continuum formulation. When such phonons reach the boundary of the atomic scale, it reflects back into the atomic region, causing unphysical heating in the atomic region. A solution for this problem, as identified by Tadmor et al. [53], is the use of the generalized Langevin equation (GLE) [54]. The GLE uses a kernel function integrated over time to represent the interaction between the boundary atoms and the missing atoms in the continuum region. However, this integral is extremely complicated except in very simple systems. The solution to this integral via numerical and analytical approximations is an active area of study [21,54–57]. It is interesting to note that Mathew et al. [57] used this method to produce a thermomechanical coupling scheme where the phonons were separated into the short wave and long wave and propagated into the continuum as heat flux and mechanical deformation, respectively. The other drawback of the QC method has been its lack of applicability to non-equilibrium molecular dynamics (NEMD) problems. Tadmor et al. [53] identify this as an improvement area and suggests that it is important to have a strong footing in the equilibrium molecular dynamics modeling first.

### 4.2. Bridging Scale Decomposition Method (BSD)

A recently developed multi-scale model that is shown to model NEMD processes is the bridging scale decomposition (BSD) method proposed by Wagner and Liu [5,17,58]. This method is conceptually very different from other partitioned domain methods because the continuum representation exists everywhere, whereas the atomic representation only exists in certain regions. The key idea here is the use of a coarse-scale, fine-scale decomposition of the displacement field. This decomposition is performed using a projection operation which, in essence, splits the total displacement field ($\mathbf{u}$) into a fine ($\mathbf{u}'$) and coarse scale ($\bar{\mathbf{u}}$) component, as shown in Equation (14).

$$\mathbf{u} = \bar{\mathbf{u}} + \mathbf{u}' \tag{14}$$

Note that the total displacement taken as the atomic displacement is obtained from MD simulation in this study, even though, in principle, this can be from DFT simulations. It is important to note here that the decomposed fine and coarse-scales are exclusive to each other. The projection of the fine-scale displacement field onto a coarse-scale shape function will be zero, and the coarse-scale component does not contain the fine-scale component, which is not representable anyway in coarse-scale shape functions. The projection operator $\mathbf{P}$ is chosen such that it minimizes the mass-weighted least square error associated with projecting the MD displacement $\mathbf{q}$ onto a finite-dimensional basis as seen in Equation (15), where the subscript $\alpha$ is given for the discrete atomic quantities, such as mass and displacement and the subscript I denotes the quantities associated with the node I. $\mathbf{w}_I$ represents a temporary nodal degree of freedom and $N_I^\alpha$ represents the shape function of node I evaluated at the location of the atom.

$$\text{Error} = \sum_\alpha m_\alpha \left( \mathbf{q}_\alpha - \sum_I N_I^\alpha \mathbf{w}_I \right) \tag{15}$$

$$\mathbf{P} = \mathbf{N}\mathbf{M}^{-1}\mathbf{N}^T\mathbf{M}_A \tag{16}$$

$$\mathbf{M}_A = \begin{pmatrix} m_1 & 0 \\ 0 & m_2 \end{pmatrix} \tag{17}$$

$$\mathbf{M} = \mathbf{N}^T \mathbf{M}_A \mathbf{N} \tag{18}$$

The projection operator $\mathbf{P}$ is defined as shown in Equation (16), where $\mathbf{M}_A$ is the diagonal MD mass matrix of the form shown in Equation (17) and $\mathbf{M}$ refers to the FE mass matrix defined as shown in Equation (18). A complimentary projector $\mathbf{Q}$ was also defined as shown in Equation (19), such that the atomic displacement can be obtained from nodal displacement as shown.

$$\mathbf{Q} = \mathbf{I} - \mathbf{N}\mathbf{M}^{-1}\mathbf{N}^T\mathbf{M}_A \tag{19}$$

The fine-scale displacement $\mathbf{u}'$ is rewritten as shown in Equation (20), and the total displacement (Equation (14)) can be rewritten as shown in Equation (21). The term containing the projection operator is referred to as the "bridging scale".

$$\mathbf{u}' = \mathbf{q} - \mathbf{Pq} \tag{20}$$

$$\mathbf{u} = \bar{\mathbf{u}} + \mathbf{q} - \mathbf{Pq} \tag{21}$$

$$\mathbf{M}_A\ddot{\mathbf{q}} = \mathbf{f} \tag{22}$$

$$\mathbf{M}\ddot{\mathbf{d}} = \mathbf{N}^T\mathbf{f}(\mathbf{u}) \tag{23}$$

Equations (22) and (23) form the coupled multi-scale equations of motion. In Equation (22), the MD displacements $\mathbf{q}$ are obtained from standard MD solvers, and the MD internal force $\mathbf{f}$ is found by minimizing potential energy functions in MD, and the nodal displacement $\mathbf{d}$ is obtained by solving Equation (23), where the other terms are known. The coupling between the two equations here is through the coarse scale force $\mathbf{N}^T\mathbf{f}(\mathbf{u})$ obtained from the MD internal force $\mathbf{f}$.

In the regions where the coarse scale mesh element is fully filled with atoms, as shown in the top part of Figure 4, the coarse scale force is obtained by interpolating the MD force. In the case where the coarse scale mesh element is not fully filled with atoms, as shown in the bottom part of Figure 4, an approximation must be made for the force otherwise determined from MD potential functions. Two methods are discussed in the BSD method to achieve this. The first method uses the Cauchy–Born rule to obtain the coarse scale internal force as shown in Equation (24), where $\mathcal{P}$ is the first Piola–Kirchoff stress and $w_q$ is the weight function associated with the quadrature point $\mathbf{X}_q$. The second method adopted by Qian et al. [5] directly applies the MD potential to each coarse scale quadrature point, as shown in Equation (25).

$$\left(\mathbf{N}^T\mathbf{f}\right)_I = -\sum_q \frac{\partial N_I}{\partial \mathbf{X}}\left(\mathbf{X}_q\right)\mathcal{P}\left(\mathbf{X}_q\right)w_q \tag{24}$$

$$(\mathbf{N}^T\mathbf{f})_I \approx -\sum_{\overline{\alpha}}\sum_{\overline{\alpha}\neq\overline{\beta}} w_{\overline{\alpha}}\frac{\partial\Phi_{\overline{\alpha}}\left(\mathbf{r}_{\overline{\alpha}\overline{\beta}}\right)}{\partial\mathbf{r}_{\overline{\alpha}\overline{\beta}}}\sum_I\left(N_I(\mathbf{X}_{\overline{\alpha}}) - N_I\left(\mathbf{X}_{\overline{\beta}}\right)\right) \tag{25}$$

In Equation (25) $\approx$ denotes the usage of a reduced set of atomic positions $\overline{\alpha}$ as quadrature points with corresponding weighting functions $w_{\overline{\alpha}}$, instead of using all the atomic positions as quadrature points. $\Phi$ represents the inter-atomic potential energy of the bond between two coarse-scale quadrature points $\overline{\alpha}$ and $\overline{\beta}$. In both cases, the idea is to use the same MD potential function as an approximation of forces between the coarse scale quadrature points. In the former method, which uses the Cauchy–Born rule, the potential energy density is used to calculate the continuum scale property, and deformation gradient $\mathbf{F}$. However, in the latter method, the MD potential function is directly applied on the coarse scale quadrature points.

The use of GLE to avoid the spurious reflection at the multi-scale boundary and to allow the FE degrees of freedom to affect the boundary atoms through the modified MD equations of motion are important features of the BSD method. At the boundary of the MD region, a region without the actual atomic degrees of freedom is defined such that it mathematically accounts for the boundary forces experienced by the boundary atoms in the MD region by employing a GLE to avoid wave reflections. In practice, the equation of motion for atoms in the MD region is modified to include the mathematically accounted for degrees of freedom. These effects are divided into two components, namely the time history kernel ($\theta(t)$), which mimics the collective behavior of the mathematically accounted-for atomic degrees of freedom, and the random force $\mathbf{R}(t)$, which accounts for the energy exchange caused due to the difference in temperature between the MD region and the GLE region. These terms appear on the right-hand side of Equation (22) as a modified MD force. For a harmonic solid, an analytical solution for the $\theta$ can be evaluated. Numerical methods for evaluating $\theta(t)$ applicable for general lattices has been developed by Wagner et al. [55] and Karpov et al. [59]

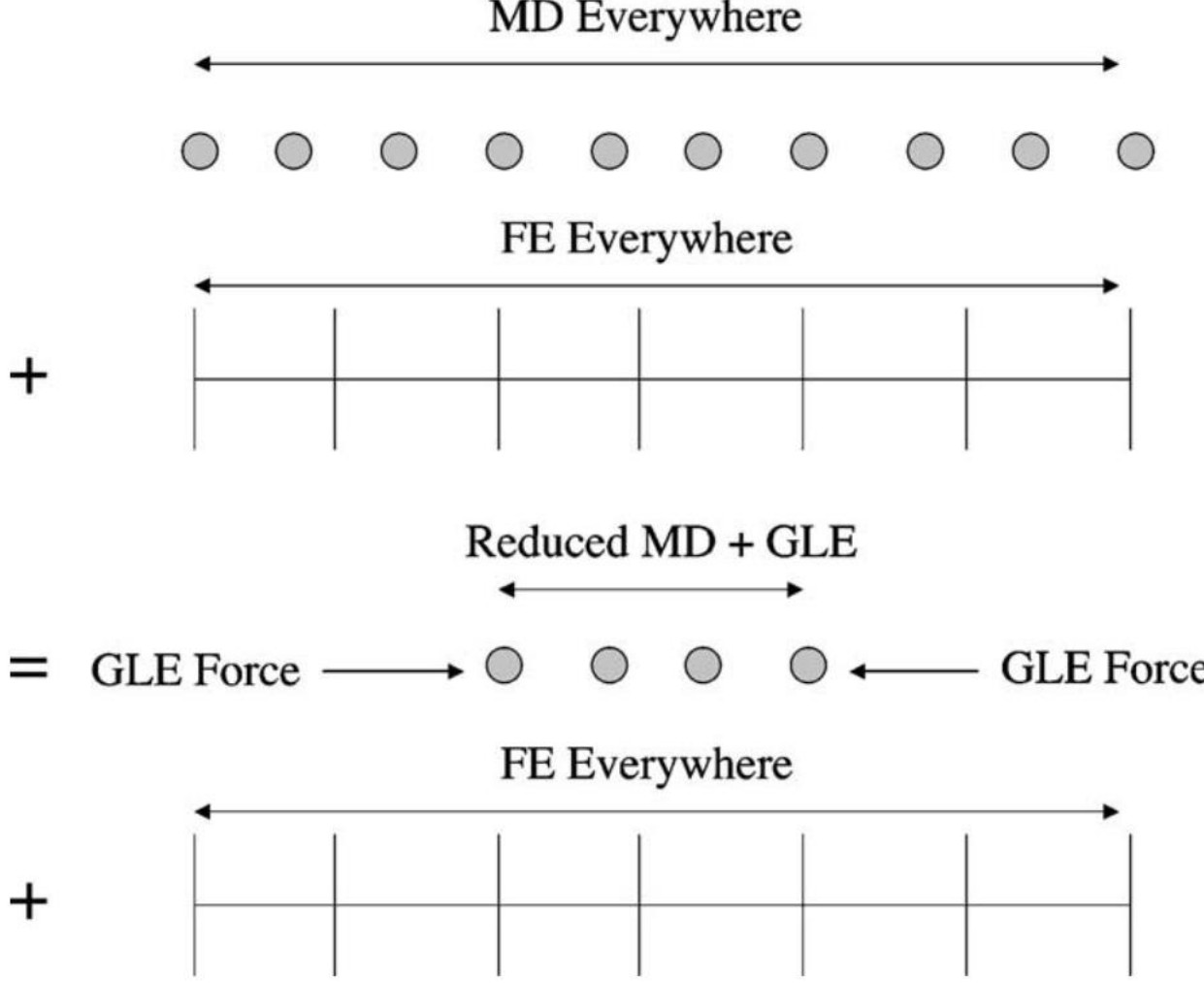

**Figure 4.** Illustration of schemes where the MD region spans the FE region fully (**top**) and partially (**bottom**). Reproduced with permission from [58].

An important feature to be noted about this method is that there is no handshake region used, eliminating the need for the one-to-one mapping of atoms and nodes as required in other methods, such as the QC method and the bridging domain (BD) method [19]. Due to the fact that the coarse scale is not meshed down to the atomic-scale as it is in other dynamic multi-scale methods such as CGDM and MAAD [12], the coarse scale simulation's time-step size does not need to be in the order of the fine scale simulation's time-step size. This feature reduces the waste in computational resources encountered when the coarse-scale simulation must be otherwise performed at the fine scale time-step size. Templeton et al. [2] demonstrated the application of the BSD method to model heat transport in NEMD simulation, where fixed temperature and fixed flux boundary conditions can be applied to MD simulation as it has been conducted on traditional FEA models.

### 4.3. Virtual Internal Bond (VIB)

Virtual internal bond (VIB) [60] is a two-scale constitutive theory to model bond breaking behavior where the micro-fracture criterion is directly built into the macro-constitutive relation through the micro-bond potential. Unlike conventional continuum mechanics, the VIB theory assumes that on the microscale, a solid consists of randomly distributed material particles. These material particles interact via virtual internal bonds. Within the framework of the hyperelasticity theory, the macro-constitutive relation is directly derived from the micro-bond potential. The micro-bond potential governs the macro-mechanical properties of the material. Since the stress–strain relationship of the VIB contains the micro-fracture mechanism, the VIB presents many advantages for simulating fracture propagation.

## 5. Thermal Coupling Methods

### 5.1. BSD Uni-Directional Temperature Coupling (BUTC)

Another method [61] that was demonstrated for representing the atomic temperature as a field in the continuum scale used the projection matrix developed in the BSD method. This method was developed so that the temperature field in the continuum region where the MD solution was unavailable could also be solved using a lattice evolution function. The bridging scale projection matrix is used to project the temperature of each atom given by Equation (26) onto the continuum temperature field vector $\mathbf{T}$ as shown in Equation (27), where $\mathbf{g}$ is the column matrix of atomic velocity squares as such that the components of the velocity related by $\mathbf{g}_i = \mathbf{v}_i^2$. $\mathbf{M}_A$ are the diagonal matrix of atomic masses.

$$\mathbf{T}_\alpha = \frac{m_\alpha}{k_B} \mathbf{v}_\alpha^2 \tag{26}$$

$$\mathbf{T} = \mathbf{P}\left(\frac{1}{k_B} \mathbf{M}_A \mathbf{g}\right) \tag{27}$$

This method then approximates the temperature of atom $\alpha$ using the FE shape function according to Equation (28), where $\theta$ are the nodal temperatures and $\mathbf{N}_I^\alpha$ is the shape function of node I evaluated at $\alpha$.

$$\mathbf{T}^\alpha = \mathbf{N}_I^\alpha \theta \tag{28}$$

Equation (29) is obtained by multiplying both sides of Equation (27) with $\mathbf{N}^T\mathbf{M}_A$ and replacing $\mathbf{T}$ and $\mathbf{P}$ with Equation (28) and Equation (16) respectively. This equation can be simplified using the equation for the FE mass matrix (Equation (18)) and written in an index notation similar to Equation (30) where subscripts I and J indicate FE nodal values and the summation is over all atoms of $\alpha$.

$$\mathbf{N^T M_A N \theta} = \frac{1}{k_B} \mathbf{N^T M_A N M^{-1} N^T M_A (M_A g)} \tag{29}$$

$$M_{IJ} \theta_J = \frac{1}{k_B} \sum_\alpha N_I(X_\alpha) m_\alpha^2 v_\alpha^2 \tag{30}$$

It is important to note here that the continuum scale nodal temperatures are obtained by using only the atomic masses and velocities from the atomic scale, and the FE shape functions from the continuum scale. The direct implication of this simplicity is that this method can be easily integrated into generic MD and FEA codes.

Though this solution works well when the FE element is filled with atoms, it is necessary to be able to find the nodal temperatures when the atoms do not completely fill the FE elements. Park et al. [61] extended the idea of using multi-scale boundary conditions using GLEs [55,59] to account for the missing atoms. In the prior work with multi-scale boundary conditions using GLEs, the effect of the missing atoms was manifested as an additional force on the boundary MD atom only. However, in this implementation, the effect of the missing atoms is propagated to the entire space. Park et al. derived expressions for the evolution function $w_n(t)$ (Equations (31) and (32)), which can be used to obtain the velocity of arbitrary atoms ($\mathbf{v_n}(t)$) outside the MD region at time t as shown in Equation (33).

$$w_n(t) = 2n \frac{J_{2n}(2\omega t)}{t} \tag{31}$$

$$\dot{w}_n(t) = -\frac{2n}{t^2} J_{2n}(2\omega t) + \frac{2n\omega}{t} (J_{2n-1}(2\omega t) - J_{2n+1}(2\omega t)) \tag{32}$$

$$\mathbf{v_n}(t) = \int_0^t \dot{w}_n(t-\tau) u_0(\tau) d\tau \tag{33}$$

To arrive at this relation, the first assumption made by Park et al. was a 1D chain of atoms interacting via a quadratic (harmonic) potential function characterized by a spring constant *k*. The term $\omega$ appearing in the evolution function is related to *k* as $\omega = \left(\frac{k}{m}\right)^{\frac{1}{2}}$ where *m* is the mass of each atom. A linear equation of motion describing the 1D atom motions is modified using a Laplace transform and a discrete Fourier transform while assuming that the initial velocity and displacement are zero. The $J_{2n}$ here is a Bessel function that appears in the solution of an inverse Laplace transform. $u_0(\tau)$ denotes the displacement of the boundary MD atom at the time interval $\tau$. The implication of this method is that with the ability to calculate the MD velocity anywhere in space, the continuum temperature can be obtained using the projection matrix anywhere outside the MD region. However, the validity of this method in higher dimensions and for more complex inter-atomic potential functions governing the atomic motion in typical MD simulations is not clearly shown. It must also be noted that the temperature equation by itself is only useful in inferring the macroscopic temperature from the underlying microscopic temperature. In other words, only up-scaling can be performed with this equation, and downscaling cannot be performed. To perform downscaling from FE to MD, it may be necessary to make use of the inverse projection matrix and couple the temperature equation with the continuum scale heat equation in some way that the temperature evolution in the coarse scale due to boundary conditions would affect the fine-scale temperature and vice versa.

*5.2. BSD Bi-Directional Temperature Coupling (BBTC)*

Some of these limitations were addressed by Wagner et al. [62]. The contributions of this work can be summarized as the introduction of downscaling using the concept of the lambda force, the modification of the upscaling procedure using row-sum lumping, and the derivation of the coupled MD/FE heat equation. The downscaling is achieved by the application of an additional drag force to the atoms in MD characterized by a multiplication factor $\lambda$, which is basically a Lagrange multiplier and is defined at the nodes and interpolated to individual atomic positions such that of $\lambda_\alpha$ which can be different for each atom. The application of the additional drag force to atom $\alpha$ is shown in Equation (34), where $\mathbf{f}_\alpha^{md}$ is the standard MD force and $\mathbf{f}^\lambda$ is the lambda force. The lambda force is given in Equation (35). As stated earlier, $\lambda$ is defined at the nodes and interpolated to the atom locations as shown in Equation (36) using the FE interpolation function $N_{I\alpha}$ for node I and $\mathcal{M}$ which is the set of all nodes whose shape functions evaluate to a non-zero value at the atom positions. In effect, they used the atoms as the quadrature points on which the integral was evaluated on the MD region.

$$m_\alpha \dot{\mathbf{v}}_\alpha = \mathbf{f}_\alpha^{md} + \mathbf{f}^\lambda \tag{34}$$

$$\mathbf{f}^\lambda = -\frac{m_\alpha}{2} \lambda_\alpha \mathbf{v}_\alpha \tag{35}$$

$$\lambda_\alpha(t) = \sum_{I \in \mathcal{M}} N_{I\alpha} \lambda_I(t) \tag{36}$$

Another feature of this work is the approximation of the projection operation using a row-sum lumping operation, such as that of the nodal temperature ($\theta_I$), which becomes an atom-to-node reduction operation as shown in Equation (37) rather than a true projection (Equation (30)). $\mathcal{A}$ is the set of all the atoms in the MD region.

$$\theta_I = \frac{1}{3k_B} \sum_{\alpha \in \mathcal{A}} \hat{N}_{I\alpha} m_\alpha \mathbf{v}_\alpha^2 \tag{37}$$

Here the coefficients $\hat{N}_{I\alpha}$ are the scaled finite element shape functions as defined in Equation (38).

$$\hat{N}_{I\alpha} \equiv \frac{N_{I\alpha} \nabla V_\alpha}{\sum_{\beta \in \mathcal{A}} N_{I\beta} \nabla V_\beta} \tag{38}$$

A heat equation for coupling the MD and FE solutions (Equation (40)) was derived, assuming the energy conservation principle (Equation (39)) to solve the change in the nodal temperature with respect to time. Here the first term on the right-hand side describes the coupled region where the MD region and FE region coexist. In this term, downscaling is incorporated through $\frac{1}{2}\mathbf{f}_\alpha^\lambda$. It is worth noticing that only half of the lambda force ($\mathbf{f}_\alpha^\lambda$) is applied on the atoms and that stems from the fact that the continuum solution solves for the total energy (kinetic and potential energy) embedded in the equipartition assumption of the Dulong–Petit law, whereas the MD solution for temperature only accounts for the kinetic energy and not the potential energy. The second and third terms on the right-hand side represent the solution in the FE region where an atomic description is absent, and these terms are the conduction terms and flux boundary condition terms, respectively, as they would routinely appear in FE problems. Here $\mathcal{N}$ is a set of all the nodes.

$$\dot{E}^{tot} = \dot{E}^{md} + \dot{E}^{fem} = 0 \tag{39}$$

$$\sum_{J \in \mathcal{N}} \left( \int_\Omega N_I N_J dV \right) \dot{\theta}_I$$

$$\begin{aligned}
= \sum_{\alpha \in \mathcal{A}} & \left( \frac{2}{3k_B} N_{I\alpha} \mathbf{v}_\alpha \cdot \left( \mathbf{f}^{md}_\alpha + \tfrac{1}{2} \mathbf{f}^\lambda_\alpha \right) + \sum_{J \in \mathcal{N}} \left( \nabla N_I \cdot \frac{\kappa}{\rho c_p} \nabla N_J \right) |_\alpha \theta_J \right) \Delta V_\alpha \\
& - \sum_{J \in \mathcal{N}} \left( \int_\Omega \left( \nabla N_I \cdot \frac{\kappa}{\rho c_p} \nabla N_J dV \right) \right) \theta_J \\
& - \int_{\Gamma_q} N_I \frac{\bar{q}_n}{\rho c_p} dA
\end{aligned} \tag{40}$$

The nodal temperatures obtained using Equations (34) and (40), however, do not include the temporal averaging discussed by Park et al. [61]. Wagner et al. [62] noticed undesirable fluctuations in the nodal temperature in the overlapped region, and it was incompatible with the smoothly varying temperature field in the FE-only region. A time-filtered version of the coupled heat equation was also developed by Wagner et al., which is not discussed here. Wagner et al. emphasizes that continuum formulations for heat transport, other than the Fourier heat law used here, may be used depending on the problem being explored, and a necessary theory or experiment should be used to ascertain the validity of the continuum heat law. This method does not provide an a priori way to determine the validity of the continuum heat law used. It is also noted that this method does not exclude the possibility of wave reflection at the multi-scale boundary since the implementation did not include the techniques used for eliminating such effects [17,55,63]. However, it is noted that the high-frequency waves accounting for the thermal energy are transported out into the continuum through the temperature coupling and that the lower-frequency waves, which account for the mechanical displacements are indistinguishable from the surrounding waves/phonons at the finite temperatures in which the study was performed. Even though a proper treatment of the multi-scale boundary condition was not performed, a layer of stationary atoms called "ghost" atoms was employed in the continuum region adjacent to the boundary atoms in the MD region so that the boundary atoms in the MD region did not experience surface effects. The "ghost" atoms are stationary because they are not included in the MD ensemble and are only used for computing the forces on the boundary atoms within the MD region.

### 5.3. BSD Spatially-Varying Thermostats (BST)

It was identified by Templeton et al. [2] that the method developed by Wagner et al. [62] was missing the ability to control the MD temperature based directly on the FE temperature or flux boundary condition. This inability was basically due to the lack of suitable thermostats that were used for the MD simulation. In this work, Gaussian isokinetic-based thermostats were chosen because they can be easily derived in non-equilibrium settings for which this multi-scale coupling method was suited. They also observed that the method used by Wagner et al. applies a thermostat force to every atom irrespective of the consistency of the temperature between the nodes and the atomic temperature of the surrounding MD region. An important contribution of this work is the introduction of spatially varying thermostats by coupling MD with FE description. This method allows for imposing Dirichlet and Neumann boundary conditions and heat source/sink in the MD simulations with complex geometries, which are otherwise very difficult to achieve in classic MD. The FE shape functions enable thermostats to provide decay rates of arbitrary smoothness along the complex geometry limited by the order of the shape functions used.

## 6. Thermomechanical Coupling Methods

Establishing a clear relationship between thermal and mechanical behaviors in a multi-scale context has been of interest to many researchers, given its applicability to many physical phenomena. A general strategy to achieve this coupling has been to decompose the atomic motion into high-frequency oscillations caused by thermal energy and low-frequency motion caused by structural deformation. It is important to note that the key is the frequency of motion and not the velocity of the atomic motion.

### 6.1. Thermo-Mechanical Equivalent Continuum Theory (TMEC)

In the TMEC theory developed by Zhou et al. [64], the atomic velocity is decomposed into a relatively high-frequency thermal oscillation part and a low-frequency structural deformation part. The thermal oscillations have been identified in the frequency range of 0.5–50 THz [65], and the low-frequency vibrations are selected based on a pre-determined cutoff frequency determined on a case-by-case basis. The actual separation is achieved through a Fourier analysis. Zhou noted that the atomic motion caused by shock waves with frequencies as high as 2 Ghz does not pose a challenge to the separation from thermal motion. However, frequencies as high as 1 THz caused by laser-induced processes have been identified as a potential problem along with other deformations which may be intimately related to the thermal motion of atoms. An important development of the TMEC theory with regard to the coupling of thermal and mechanical processes is the development of the inertial forces term that appears in both the structural deformation equation and the heat equation. In the structural deformation equation, the inertial force induced by the thermal oscillations acts like an invisible external force. In the heat equation, the inertial force induced by the structural deformation acts like a heat source.

### 6.2. Filter and Restitution Method (FR)

The idea of separating the phonon spectrum into a thermal part and a deformation part was explored by Mathew et al. [57] to couple thermal and mechanical behaviors from an atomic to the continuum scale. The simulation domain was divided into a fully atomic region ($A$) and a continuum region ($C$), separated by an interface region ($I$). The interface region is subdivided into the "filter and restitution" region ($R + F$), and in two regions $A(C)$ and $C(A)$ the mechanical coupling was performed by imposing multi-scale boundary conditions. The $R + F$ region takes care of separating and blocking out the high-frequency part of the phonon spectrum from entering the continuum region through the "filtering" process ($F$). This is conducted through a frequency-dependent damping term in the GLE, which describes the motion of atoms in this region. This damping function uses a memory kernel $G$. This idea has been explored earlier in the BSD method [25,59] and is also known as the lattice dynamics Green's function in solid-state physics. The "restitution" process ($R$) serves the purpose of returning energy to the $A$ region to maintain energy conservation. The $R$ process also serves the purpose of heat flux coupling between the $A$ region and the thermal component of $C$. The $R$ process is achieved through a random force term in the GLE. This random force is non-zero in $R + F$ and zero elsewhere and also uses the Wiener–Khinchin theorem [66] to return the energy extracted by the $F$ process into the phonon modes from which it was extracted. However, the exact details of how the force is augmented based on the position within the $R + F$ region is not clear. The mechanical coupling is performed by imposing the displacement of the continuum nodes based on minimizing the least squared difference between the atomic and continuum displacement. This method can be thought of as a BSD method derivative. This method was demonstrated for 1D problems in thermal equilibrium and in non-equilibrium.

### 6.3. Atom-Continuum Coupled Model (ACC)

The recently developed atom-continuum coupled model (ACC) [67] that couples thermo-mechanical behavior in micro-nano scales uses a time averaging scheme to approximate the separation of thermal vibration parts of the phonon spectrum from the structural deformation part instead of using the Fourier transform approach [64] or the memory kernel approach [57]. This idea assumes that in the absence of mechanical deformation and under equilibrium temperature conditions, atoms vibrating about an equilibrium position are reduced to stationary atoms when the displacements are time averaged over time scales larger than the thermal vibration frequencies. Then, the atomic motion observed with time averaging is classified as the structural deformation motion. This method uses a representative volume element (RVE) similar to the approach used in the QC method. In some ways, this method can be considered a QC method derivative. Surface effects are avoided using an extended representative volume element (ERVE), an approach that can be compared to the "ghost atoms" [2]. However, there is no clear indication about handling spurious reflections at the multi-scale boundary. This method also uses a 1D problem to demonstrate thermo-mechanical coupling.

### 6.4. Wavelet-Based Denoising (WBD)

The idea of considering the problem of the thermomechanical decomposition of atomic velocity as a denoising problem in signal processing was introduced by To et al. [68]. They compared a wavelet-based method with a Fourier-based method called the Weiner method and moved average methods for denoising spatial velocity data. The smooth, noiseless velocity data were obtained from four test signals as well as a snapshot of an MD simulation of crack propagation taken at 1 ns. Both the analytical data and MD velocity data were corrupted by thermal velocity data obtained from MD simulations where the velocities followed the Maxwell–Boltzmann distribution for prescribed temperatures.

### 6.5. Finite Temperatures Using Spatial Filters (SF)

The problem of the spurious wave reflection in the AtC methods has been a barrier for finite temperature simulations. The application of the coupling methods to handle non-equilibrium processes, such as heat conduction, is limited. The idea of using spatial filters (SF) with least square minimization as a new scale transfer operator was proposed by Ramisetti, et al. [69] for thermos-mechanical problems to couple MD with FE at finite temperatures. The mismatch in the dispersion relations between the continuum and atomistic models leads to unwanted mesh vibrations, which are illustrated using a standard least square coupling formulation. This method was developed to use selective spatial filters for damping the wavelength modes for coupling atomistic and continuum models at finite temperatures. The restitution force from the generalized Langevin equation was modified to perform a two-way thermal coupling between the two models. The application of the proposed method has been validated by a high-velocity impact test in two-dimensional space.

### 6.6. Extended Irving-Kirkwood Statistical Mechanical (EIK)

The extended Irving–Kirkwood (EIK) method was introduced by Chen et al. [70] for the modeling and simulation of crystalline methods. The formulation extends the Irving–Kirkwood statistical mechanical procedure for deriving transport equations and fluxes for homogenized molecular systems to that for polyatomic crystalline materials by employing a concurrent two-level description of the structure and dynamics of the crystals. A multiscale representation of conservation laws was formulated as a direct consequence of Newton's second law in terms of the instantaneous expressions of unit cell-averaged quantities using the mathematical theory of distributions. Conservation equations were derived in terms of the instantaneous expressions of cell-averaged quantities using the mathematical theory of distributions. Fluxes can be then obtained to quantify the flow of

conserved quantities across the lattice cells as well as those that flow back and forth within the cells, as direct consequences of the local density definition and Newton's second law.

## 7. Conclusive Remarks

The richness and diversity of this subject make it a non-trivial task to review the relevant literature. This review is intended to provide a guideline for researchers beginning to explore AtC methods from their original motivations and formulation to recent improvements. Table A1 (Appendix A). summarizes the AtC methods reviewed in this article. Molecular dynamics, solid mechanics, numerical methods, and materials science have been identified as the knowledge areas where competency is necessary to make appreciable contributions in multi-scale modeling. Even though several reviews and textbooks that cover these topics and some concepts in mechanics, such as the virial theorem (VT) and Cauchy–Born rule, and mathematical tools, such as Gaussian thermostats and the generalized Langevin equation, show up frequently in the literature and the importance of understanding the fundamental idea behind such specialized topics cannot be disregarded. The motivation for multi-scale modeling through examples of applications is first introduced, and then the taxonomy of the methods is presented. This review has also attempted to portray the gradual evolution of the multi-scale coupling methods from the simple idea of the interpretation of atomic scale quantities in the continuum sense [26] to highly advanced mechanisms for concurrently mapping the atomic and continuum scale quantities onto each other [17,18]. Earlier methods focused on capturing the non-local nature of the atomic scale properties and accurately mapping to a local form typical of the continuum formulation. The focus then shifted to avoiding spurious reflections in the multi-scale boundary. The current focus now seems to be shifting towards the application of the already mature multi-scale techniques to material problems where multi-physics treatments become necessary. Eight methods that couple material deformation, four methods that couple thermal behavior, and three methods that couple thermo-mechanical behavior have been reviewed. An introductory section is presented for each class of methods, where key concepts that are common to all methods are identified and discussed. The key contribution of each method is presented in a concise manner in Table A1. An attempt to capture the evolution history is given in Figure 5. It must be noted that some methods have been assigned a name and acronym for the sake of convenience and may not capture the full identity of the original method. Hence, a reference to the papers is also provided in Table A. In Figure 5, some methods are related to other methods, using dotted lines to represent the obvious connection between the methodologies even if the relationship is not explicitly stated in the derived method. For example, the FR method has many characteristic features of the BSD method and has extended the ideas originally developed in the TMEC method.

However, this is by no means an exhaustive account of the developments in this field. Several reviews [21,71–75] have captured the sheer amount of ongoing research in this nascent field and have identified the application areas and challenges that lay ahead [1,76,77]. The current multi-scale methods for coupling thermal and mechanical phenomena have matured enough to provide a solid basis for building robust multi-scale, multi-physics models which have the potential to solve thermo-mechanical problems in several disciplines that are yet to be elucidated.

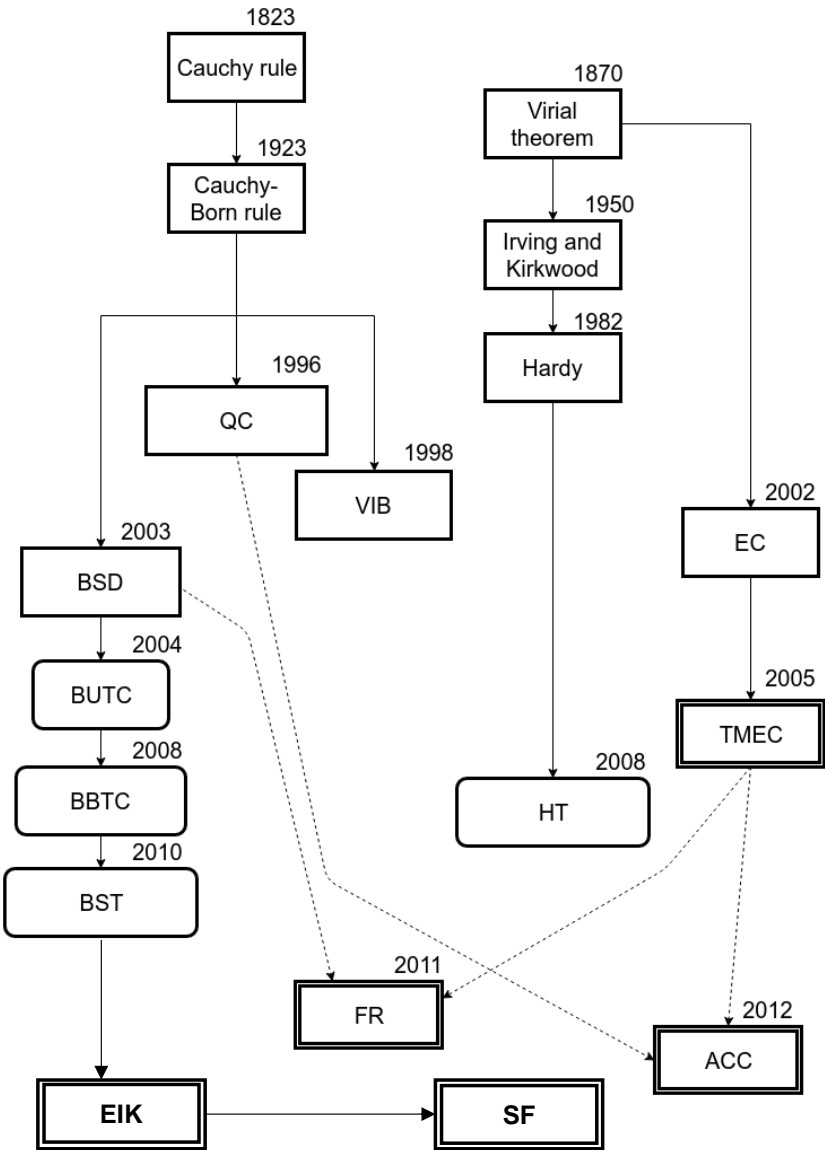

**Figure 5.** A graphical representation of the evolution of multiscale methods for modelling thermal and mechanical behavior. Rectangles with single border and sharp corners are used for models that only consider mechanical behavior. Rectangles with rounded corners denote models that couple thermal behavior only and double borders are used for models that couple thermal and mechanical behavior. Dotted lines connecting some methods represent the obvious connection between the methodologies even if the relation is not explicitly stated in the derived method.

**Author Contributions:** Conceptualization, Y.L.; methodology, S.T. and T.J.Z.; writing—original draft preparation, Y.L. and S.T.; writing—review and editing, Y.L. and T.J.Z.; supervision, Y.L. All authors have read and agreed to the published version of the manuscript.

**Funding:** This research received no external funding.

**Data Availability Statement:** No new data were created or analyzed in this study. Data sharing is not applicable to this article.

**Conflicts of Interest:** The authors declare no conflict of interest.

## Appendix A. Tables

**Table A1.** Summary of the AtC methods reviewed in this article.

| Method | Features |
|---|---|
| VT [26] | Introduced the ability to map discrete atomic quantities to a continuum representation of stress as a point-wise function using summation over sub ensembles. |
| CB | Introduced a method to map uniform deformation at macroscale to atomic lattice deformation. |
| IK [31] | Summation over sub ensembles in VT is replaced by density of atomic functions in conjunction with a Dirac delta function as the localization function. |
| EC [41] | A purely mechanical theory which defines work-conjugate stress and deformation fields from non-local MD system. |
| HM [35] | Introduced localization functions with finite range (compact support) instead of using just the Dirac delta function as in IK. |
| QC [18] | Introduced the idea of deriving constitutive parameter for FE nodes (rep atoms) from a small set of atoms (crystallite) based on CB. |
| BSD [17] | Introduced the idea of decomposing total displacement can be MD displacement) into coarse and fine scale components which can and cannot be projected to FE shape functions respectively. |
| VIB [60] | Introduced the idea of filling continuum with virtual material points which are not atoms but obey a pseudo potential energy function based on experimental data. |
| HT [30] | Uses the relative atomic velocity derived in Hardy method to define a weighted average temperature. |
| BUTC [61] | Uses the projection matrix derived in BSD to project atomic temperature to FE shape functions, i.e., up-scaling only. |
| BBTC [62] | Extended BUTC by implemented down-scaling and introduced row-sum lumping as an approximation for the projection method used in BUTC and BSD. |
| BST [2] | Introduced the ability to prescribe spatially varying initial conditions, Neumann and Dirichlet boundary conditions to the MD-FE coupled material model using Gaussian isokinetic thermostats. |
| TMEC [64] | Introduced the idea of decomposing atomic motion into high frequency thermal motion and low frequency structural deformation motion using Fourier analysis. |
| FS [57] | Demonstrated 1D examples for thermo-mechanical coupling model derived from BSD and TMEC methods. |
| ACC [67] | Demonstrated 1D examples for thermo-mechanical coupling model derived from QC and TMEC methods but uses an averaging scheme to approximate the Fourier analysis method used in TMEC. |
| SF [69] | Demonstrated 2D example of a high-velocity impact test using selective spatial filters for damping the wavelength modes for coupling atomistic and continuum models at finite temperatures. |
| EIK [70] | Demonstrated 2D examples of fluxes that can be then obtained to quantify the flow of conserved quantities across the lattice cells as well as those that flow back and forth within the cells. |

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
