# Peer review of "Concurrent AtC Multiscale Modeling of Material Coupled Thermo-Mechanical Behaviors: A Review"

_2673-4109, doi:10.3390/civileng3040057_

Round 1

Reviewer 1 Report

1. My major concern is the scientific frontiers of this review. Many references are between 2000-2010 and even before 2000. The authors may try to include more state-of-art researches in the review to show their close attentions on the most advancing progress in the direction.

2. Figure 5 is interesting. The authors can add proper references to each representative model. The authors did such in the Appendix A, but it can be better to incorporate these references in the evolution tree. Besides, is there any further updates after 2012?

3. As a review paper, the model development is essential and the authors have shown a clear history. However, a nomenclature is suggested to show more clearly the meanings of the symbols common seen in this area. 

4. In the review of thermostats (Sec 3.1),  the authors can cover the most advancing studies on NVT ensembles, for example, "https://doi.org/10.1002/er.8234" and "https://doi.org/10.3390/en14227724".

5. As a review paper, it is a always a challenge to conclude certain informative remarks for future studies. The authors can enrich the Section 7 with more such information, better in point-by-point discussion. 

Author Response

Dear CivilEng Editor,

Thank you for considering our paper for publication after we have revised it according to the reviewers’ comments. We would also like to thank the reviewers for pointing out many places where the paper could be improved. We have responded to all their comments, as detailed below. Our responses are indented with respect to the reviewers’ comments. The changes we made to our revised manuscript were extensive.  and the Track Changes would give too messy of a manuscript. The revised manuscript used “track changes” function for all edits where changes have been made.

Reviewer #1:  

  1. My major concern is the scientific frontiers of this review. Many references are between 2000-2010 and even before 2000. The authors may try to include more state-of-art researches in the review to show their close attentions on the most advancing progress in the direction.

Thanks for the suggestion. We have gone through the paper carefully. A few more recent publications have been added to include more state-of-art research in the review. It is noteworthy that the major innovations in AtC method development were between 2000 - 2015. While some minor progresses were developed, the original ideas were seldomly made after that booming period.

  1. Figure 5 is interesting. The authors can add proper references to each representative model. The authors did such in the Appendix A, but it can be better to incorporate these references in the evolution tree. Besides, is there any further updates after 2012?

It is not necessary to add reference to the tree. Because the authors want to keep Fig 5 neat and clear. Relevant references have already been added in Appendix A. It is noteworthy that the major innovations in AtC method development were between 2000 - 2015. While some minor progresses were developed, the original ideas were seldomly made after that booming period. That’s why our review focused on publications before 2015.

  1. As a review paper, the model development is essential, and the authors have shown a clear history. However, a nomenclature is suggested to show more clearly the meanings of the symbols common seen in this area. 

Nomenclature is usually used for a consistent derivation of a series of equations. However, this is a review paper that covers a broad range of scales and variables that have no commonly shared variables. Thus, it doesn’t make it more clear presentation to use nomenclature.

  1. In the review of thermostats (Sec 3.1), the authors can cover the most advancing studies on NVT ensembles, for example, "https://doi.org/10.1002/er.8234" and "https://doi.org/10.3390/en14227724".

Thanks for the suggestions. These two references have been cited in our paper.

  1. As a review paper, it is always a challenge to conclude certain informative remarks for future studies. The authors can enrich the Section 7 with more such information, better in point-by-point discussion. 

      Thanks for the suggestion. As a comprehensive review of AtC area, we have already provided specific summaries of each area in the conclusive remarks section.

Reviewer 2 Report

Unfortunately, this paper has no chance at this stage even if a major revision is suggested by the Reviewer. While I tried to prepare a careful review that could be useful for a major revision for this paper to be published at some point, it is rather pointless at this stage. 

While the topic is very interesting, an the authors have a made an effort to collect some literature, the authors make misleading claims, the literature has not connection with each other and the topic at times, and generally the whole presentation is poor.

By no means, I can suggest publication of this review article at this stage, despite my willingness to provide a positive feedback for the authors, which could be used for a major revision. Hence, I suggest that this paper be rejected.

1) abstract: It needs to be concise, focused and well-written. The current version falls behind.

2) Introduction: 

a) Authors refer to “ Multi-scale material model … time and length scales and relate them”. Do the authors know any method that is able to couple the time scales of the simulation? Here, it would be good to give a few examples.

b) Authors mention that “… multi-scale methods specialise in .. atomic scale (angstrom scale) and the microscopic continuum scale…” Can the authors name such method?? What is the “microscopic continuum method”???

c) I would divide the class of multiscale simulations in the categories the the authors are mentioning  in the intro.

d) Authors make a lot of general claims in their introduction without any reference to the literature until ref. [1] appears later.

e) There is no ‘mesoscale science term’

f) MD and DFT is not the same thing. Also, what kind of DFT, classical or quantum?

g) ….

I could go over and over every detail, but it is not possible to do so in this review. I simply urge the authors to prepare a new manuscript from scratch.

Author Response

Dear CivilEng Editor,

Thank you for considering our paper for publication after we have revised it according to the reviewers’ comments. We would also like to thank the reviewers for pointing out many places where the paper could be improved. We have responded to all their comments, as detailed below. Our responses are indented with respect to the reviewers’ comments. The changes we made to our revised manuscript were extensive.  and the Track Changes would give too messy of a manuscript. The revised manuscript used “track changes” function for all edits where changes have been made.

Reviewer #2:

Unfortunately, this paper has no chance at this stage even if a major revision is suggested by the Reviewer. While I tried to prepare a careful review that could be useful for a major revision for this paper to be published at some point, it is rather pointless at this stage. While the topic is very interesting, the authors have a made an effort to collect some literature, the authors make misleading claims, the literature has not connection with each other and the topic at times, and generally the whole presentation is poor. By no means, I can suggest publication of this review article at this stage, despite my willingness to provide positive feedback for the authors, which could be used for a major revision. Hence, I suggest that this paper be rejected.

Specific and detailed comments are more helpful than general critiques. It is suggested to provide constructive suggestions instead. The following is a response in detail. 

1) abstract: It needs to be concise, focused and well-written. The current version falls behind.

Thanks for the suggestions. We made some editing according to the suggestions.  Now it is more concise.

2) Introduction: 

  1. a) Authors refer to “ Multi-scale material model … time and length scales and relate them”. Do the authors know any method that is able to couple the time scales of the simulation? Here, it would be good to give a few examples.

Thanks for the good catch. “Time scale” is a typo. It has been deleted.

  1. b) Authors mention that “… multi-scale methods specialise in .. atomic scale (angstrom scale) and the microscopic continuum scale…” Can the authors name such method?? What is the “microscopic continuum method”???

We never specifically say “microscopic continuum method.” The sentence here means that multiscale methods link the atomic scale to continuum scale (sub-micrometer). We have clarified it in the revised manuscript by using “continuum scale”.   

  1. c) I would divide the class of multiscale simulations in the categories the the authors are mentioning  in the intro.

This is not necessary. The paper has already articulated the multiscale simulations in the categories well.

  1. d) Authors make a lot of general claims in their introduction without any reference to the literature until ref. [1] appears later.

General claims in the introduction are to give an overview. References have already been provided in the main body of this paper. 

  1. e) There is no ‘mesoscale science term’

We have already provided citations referring to mesoscale science. Please check it.

  1. f) MD and DFT is not the same thing. Also, what kind of DFT, classical or quantum?

General claims of DFT and MD methods in the introduction are to give an overview. There is no need to specify what kind of DFT, classical or quantum in the Introduction section. 

  1. g) ….

I could go over and over every detail, but it is not possible to do so in this review. I simply urge the authors to prepare a new manuscript from scratch.

      We have gone through the paper carefully, identifying repetitive text that could be eliminated. We have made some changes, including the abstract, to emphasize better the message of the paper, and shortened and better focus the introduction.

Reviewer 3 Report

The manuscript (civileng-1913143), Concurrent AtC Multiscale Modeling of Material Coupled Thermo-Mechanical Behaviors: A Review, shows an interesting review of the AtC Multiscale Modeling. Authors presented quite comprehensive analysis in this work. Some minor comments would like to provide here and hope it can improve the manuscript before further consideration, shown as following - 

1. The abstract should edit that due to repeating sentences. 

2. In Figure 5, A graphical representation of the evolution of multiscale methods for modelling thermal and mechanical behavior, authors show the review of the evolution of multiscale methods for modelling thermal and mechanical behavior until 2012, does that mean there is no further development after 2012? If there is no development needed or demanded after 2012, then is it still a need to review here? Please add further multiscale methods development after 2012 until present. 

3. There are several multiscale simulation tools now common for researchers, e.g., COMSOL and etc..., what would be the multiscale methods for modelling thermal and mechanical behavior? What would be the Pros and Cons for those multiscale simulation tools regarding their methods. Please provide a simple benchmark Table to compare those multiscale simulation tools and their corresponding multiscale methods with detailed analysis. 

Due to the above comments, this referee would like to put the manuscript status as "Major Revision" in the current phase.

Author Response

Dear CivilEng Editor,

Thank you for considering our paper for publication after we have revised it according to the reviewers’ comments. We would also like to thank the reviewers for pointing out many places where the paper could be improved. We have responded to all their comments, as detailed below. Our responses are indented with respect to the reviewers’ comments. The changes we made to our revised manuscript were extensive.  and the Track Changes would give too messy of a manuscript. The revised manuscript used “track changes” function for all edits where changes have been made.

Reviewer #3:

The manuscript (civileng-1913143), Concurrent AtC Multiscale Modeling of Material Coupled Thermo-Mechanical Behaviors: A Review, shows an interesting review of the AtC Multiscale Modeling. Authors presented quite comprehensive analysis in this work. Some minor comments would like to provide here and hope it can improve the manuscript before further consideration, shown as following - 

  1. The abstract should edit that due to repeating sentences. 

We have gone through the paper carefully, identifying repetitive text that could be eliminated. We did find a significant amount of these sentences, as the reviewer mentioned, and have eliminated them.

  1. In Figure 5, A graphical representation of the evolution of multiscale methods for modelling thermal and mechanical behavior, authors show the review of the evolution of multiscale methods for modelling thermal and mechanical behavior until 2012, does that mean there is no further development after 2012? If there is no development needed or demanded after 2012, then is it still a need to review here? Please add further multiscale methods development after 2012 until present. 

Thanks for the comments. It is noteworthy that the major innovations in AtC method development were between 2000 - 2015. Although still a lot of problems remain unsolved, the major funding sources have switched to other fields since then. While some minor progresses were developed, the original ideas were seldomly made after that booming period although a lot of problems remain unsolved. That’s why our review focused on publications before 2015. In the revised version, we have added some recent publications to show the most up-to-date progress in AtC multiscale modeling area.

  1. There are several multiscale simulation tools now common for researchers, e.g., COMSOL and etc..., what would be the multiscale methods for modelling thermal and mechanical behavior? What would be the Pros and Cons for those multiscale simulation tools regarding their methods. Please provide a simple benchmark Table to compare those multiscale simulation tools and their corresponding multiscale methods with detailed analysis. 

Due to the above comments, this referee would like to put the manuscript status as "Major Revision" in the current phase.

         It is noteworthy that COMSOL is NOT a multiscale modeling tool. Instead, it is a general finite element simulation software that only does Multiphysics modeling. Continuum scale thermal and mechanical behaviors can be simulated using COMSOL. A benchmark table to compare those multiscale simulation tools is not necessary. Because these AtC methods only handle a specific area and are thus not comparable to each other.

Reviewer 4 Report

AtC multi-scale material models allow simulating the behavior of materials at nano scale level. This paper focus reviews the state of the art of atomic to continuum (AtC) methods for simulating the thermo-mechanical behavior of materials. To this end, methods that couple mechanical behavior, thermal behavior and thermo-mechanical behavior are reviewed with the aim to provide an evolutionary perspective of the thermo-mechanical coupling methods.

The authors are asked to answer the following remarks in order to improve some aspects of the manuscript,

1.  Introduction section. The authors must clarify whether the different methods reviewed can be applied to solids, liquids and gases.

2.  Introduction section. There are no references between lines 51-64, pleas elaborate.

3.  Introduction section. Line 154: “The focus of this paper ..” Which paper? Please clarify.

4.  Introduction section. Please add a paragraph at the end of this section further stressing the contributions and applicability of this paper.

5.  All symbols in the equations must be defined.

6.  All equations must be referred.

7.  A critical section is required, which could include a Table summarizing the main characteristics, applicability, advantages and disadvantages of each method. In addition, a subsection detailing the research needs in this field would be appreciated.

 I hope a review of the paper based on these remarks can help to improve the quality of the paper.

Author Response

Dear CivilEng Editor,

Thank you for considering our paper for publication after we have revised it according to the reviewers’ comments. We would also like to thank the reviewers for pointing out many places where the paper could be improved. We have responded to all their comments, as detailed below. Our responses are indented with respect to the reviewers’ comments. The changes we made to our revised manuscript were extensive.  and the Track Changes would give too messy of a manuscript. The revised manuscript used “track changes” function for all edits where changes have been made.

Reviewer #4:

AtC multi-scale material models allow simulating the behavior of materials at nano scale level. This paper focus reviews the state of the art of atomic to continuum (AtC) methods for simulating the thermo-mechanical behavior of materials. To this end, methods that couple mechanical behavior, thermal behavior and thermo-mechanical behavior are reviewed with the aim to provide an evolutionary perspective of the thermo-mechanical coupling methods.

The authors are asked to answer the following remarks in order to improve some aspects of the manuscript,

  1. Introduction section. The authors must clarify whether the different methods reviewed can be applied to solids, liquids and gases.

Generally, AtC methods are not restricted by application domain’s physical state, such as solid, liquid or gas. In other words, AtC methods are applicable to all three states of matters. Therefore, there is no need to clarify physical states applicable.

  1. Introduction section. There are no references between lines 51-64, pleas elaborate.

Yes, we have added some references in the introduction section.

  1. Introduction section. Line 154: “The focus of this paper ..” Which paper? Please clarify.

Revised. Now it is “The focus of this review …”

  1. Introduction section. Please add a paragraph at the end of this section further stressing the contributions and applicability of this paper.

A major elaboration of the contributions and applicability of this paper has been provided in the “conclusive remarks” section. Thanks for the suggestion. We have added a couple of sentences in the Introduction section stating the contributions and applicability of this paper.

  1. All symbols in the equations must be defined.

Yes, we have already made notes for all the symbols in the equations.

  1. All equations must be referred.

Yes, all equations have already been referred.

  1. A critical section is required, which could include a Table summarizing the main characteristics, applicability, advantages and disadvantages of each method. In addition, a subsection detailing the research needs in this field would be appreciated.

 I hope a review of the paper based on these remarks can help to improve the quality of the paper.

        The research needs statement has been provided in the introduction. A Table summarizing the main characteristics, and applicability of each method has already been reviewed in detail in each section and summarized in the Appendix.

Round 2

Reviewer 2 Report

I am really sorry. I cannot recommend publication of this manuscript. As a review it only represents a superficial overview of the field with very general descriptions, inadequately linked, which hardly will be a source of information for anybody working in computational research.

Author Response

The following is the only comment made by Reviewer #2 in round #2.

-------------------------------------------------------------------
Check lines 225-231 (there are two "Section three" and no "Section four").
-------------------------------------------------------------------

Thanks for the comments. We have revised the section number in lines 225-231. Now there is only one Section three and one Section four in the sentence. The following section numbers are all correct.

Reviewer 3 Report

Authors have replied to this referee in detail. No further comments from this referee. 

Author Response

Thank you for considering our paper for publication after we have revised it according to the reviewers’ comments.

Reviewer 4 Report

The authors have replied my questions

Author Response

Thank you for considering our paper for publication after we have revised it according to the reviewers’ comments